# Characterization of Nitrogen-Doped TiO$_2$ Films Prepared by Arc Ion Plating without Substrate Heating in Various N$_2$/O$_2$ Gas Mixture Ratios

Hsing-Yu Wu [1,2], Wen-Chun Huang [3], Jyh-Liang Wang [4], Guo-Yu Yu [5], Yung-Shin Sun [3,*] and Jin-Cherng Hsu [3,6,*]

1 System Manufacturing Center, National Chung-Shan Institute of Science and Technology, New Taipei City 237209, Taiwan
2 Center for Astronomical Physics and Engineering, Department of Optics and Photonics, National Central University, Taoyuan City 320317, Taiwan
3 Department of Physics, Fu Jen Catholic University, New Taipei City 242062, Taiwan
4 Department of Electronic Engineering, Ming Chi University of Technology, New Taipei City 24301, Taiwan
5 Department of Engineering and Technology, School of Computing and Engineering, University of Huddersfield, Queensgate, Huddersfield HD1 3DH, UK
6 Graduate Institute of Applied Science and Engineering, Fu Jen Catholic University, New Taipei City 242062, Taiwan
* Correspondence: 089957@mail.fju.edu.tw (Y.-S.S.); 054326@mail.fju.edu.tw (J.-C.H.)

**Abstract:** Nitrogen-doped TiO$_2$ films exhibit good photocatalytic ability in the visible (VIS) light region. This study reports the fabrication of these films using arc ion plating (AIP) in different ratios of nitrogen partial pressure (P$_{N2}$) to oxygen partial pressure (P$_{O2}$) without substrate heating and/or applied bias. This approach allows a significant broadening of the range of possible substrates to be used. X-ray diffraction (XRD) patterns indicate that these films deposited at room temperature are amorphous, and surface electron microscope (SEM) and atomic force microscope (AFM) images show that they have rough surfaces. Their transmittance and optical properties are measured with a spectrometer and ellipsometer, respectively. In addition, the bandgap energies of these amorphous films are derived by the ellipsometer from the Tauc–Lorentz (TL) model. The results indicate that the N-doped TiO$_2$ film with a P$_{N2}$/P$_{O2}$ ratio of 1/4 attains the narrowest bandgap and the highest absorbance in the visible region. It can be attributed to the prominent Ti–N peaks observed in the sample's Ti and N X-ray photoelectron spectroscopy (XPS) spectra. In addition, verified with the methylene blue (MB) test, this sample exhibits the best photocatalytic performance for its narrowest energy gap.

**Keywords:** nitrogen-doped TiO$_2$; arc ion plating; TiO$_2$ photocatalysis; bandgap narrowing; Tauc–Lorentz model

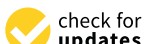



## 1. Introduction

Photocatalysis refers to a photo-activated process where pairs of chemical reactions occur to decompose organic molecules. It is widely used in various applications, including air and water purification, self-cleaning, deodorizing, sterilization, and sustainable regeneration [1–4]. The most commonly used photocatalysts are semiconductors such as ZnO, WO$_3$, SnO$_2$, and TiO$_2$ [5–9]. The photocatalytic performance of a semiconductor depends on its electronic band structure and bandgap energy, with, in general, a narrower gap giving a higher catalytic capability [10]. Among these materials, TiO$_2$ has been extensively investigated due to its low cost, photostability, redox efficiency, nontoxicity, and availability [11]. Through strong redox processes, TiO$_2$ is able to attain photocatalytic decomposition of organic pollutants in water and air [12]. TiO$_2$ has three stable crystal phases: tetragonal anatase, rutile, and orthorhombic brookite. The anatase phase possesses the best

photocatalytic efficiency because its surface area is larger than the rutile one, resulting in larger reaction areas for surface water molecules [13]. With a bandgap of around 3.2 eV for anatase $TiO_2$, an ultraviolet (UV) light at about 385 nm releases the electrons on its surface to form positively charged electron holes. Then hydroxyl groups (-OH) released by nearby water molecules are oxidized and become very active hydroxyl radicals. As an organic molecule comes across these radicals, its electrons will be captured, leading to the breakdown of its bonds and subsequent disintegration into water and carbon dioxide [13].

However, the applications of $TiO_2$ are limited by some shortcomings. The most significant limitation is that it requires UV excitation with a wavelength of less than 385 nm. Given that UV light is only about 5%~8% of the total solar intensity, the photocatalytic efficiency and energy conversion of $TiO_2$ under solar radiation, especially visible (VIS) light, are very low. To increase its absorption of VIS light, many studies have been conducted by changing the composition and microstructure of $TiO_2$ or its complexes [14–16]. Another limitation is that commercially available $TiO_2$ products are often in powders, easily lost during use, and possibly cause health problems from inhaling. As a result, $TiO_2$ thin films have been widely investigated to serve as an alternative to powder-based photocatalysts. Many techniques have been applied to the deposition of $TiO_2$ films, including sol–gel, different kinds of sputtering (dc magnetron, ion beam), different kinds of vapor deposition (chemical, physical, metal-organic chemical), electrochemical deposition, electrophoretic deposition, pulsed laser deposition, hydrothermal synthesis and arc ion plating (AIP) [17–21].

To enhance the VIS light absorption of $TiO_2$, doping a third element, such as F, C, S, P, and N, to replace oxygen atoms in $TiO_2$ has been proven to reduce its bandgap effectively [22–25]. Asahi et al. calculated the density of states of such nonmetal doping in anatase $TiO_2$ crystals. They found that the N-doping was the most efficient due to the contribution of its p states to reducing the bandgap via mixing with O 2p states [22]. Although S-doping exhibited a similar bandgap narrowing, it was not easy to incorporate it into $TiO_2$ crystal because of its large ionic radius [22]. In addition to bandgap narrowing, impurity energy levels and oxygen vacancies also follow the substitutional doping of N in $TiO_2$ [26]. Isolated impurity levels above the valence band are formed as oxygen atoms are replaced with nitrogen atoms, leading to electronic excitation under VIS light illumination [27,28]. Therefore, oxygen vacancies in the grain boundaries of N-doped $TiO_2$ are crucial to enhancing VIS light activity and blocking re-oxidation [29].

Given the applicability of $TiO_2$ in films and its enhanced photocatalytic performance in VIS light region after N-doping, N-doped $TiO_2$ films have been widely investigated with various fabrication methods. Hsu et al. prepared N-doped $TiO_2$ films using ion beam sputtering in a mixed atmosphere of $NH_3$ and $O_2$ [26]. X-ray photoelectron spectroscopy analysis of these films showed the presence of six ions: $N^{3-}$, $N^{2-}$, $N^{-}$, $N^{+}$, $N^{2+}$, and $N^{3+}$, and at an $NH_3/O_2$ ratio of 3, the bandgap of N-doped $TiO_2$ reduced to around 2.54 eV [26]. By dc magnetron sputtering, Lindgren et al. fabricated nano-crystalline porous N-doped $TiO_2$ films in plasma of mixed gases of Ar, $O_2$, and $N_2$ [30]. These films exhibited VIS light absorption in the 400~535 nm wavelength with an increased photo-induced current compared to un-doped $TiO_2$ [30]. Chemical vapor deposition was used to deposit N-doped $TiO_2$ thin films with $N_2O$ as the nitrogen dopant source [31]. Compared to pure $TiO_2$, these N-doped films attained a relatively narrow optical bandgap, resulting in enhanced VIS light-induced photocatalysis [31]. Via a single-step sol–gel synthesis, Powell et al. prepared N-doped $TiO_2$ thin films, which exhibited photo-induced super hydrophilicity under filtered white light conditions [32]. Interstitial rather than substitutional nitrogen contents led to significant red-shift in the bandgap, making these films excellent VIS light photocatalysts [32]. Among these methods, arc ion plating (AIP) is particularly interesting because anatase $TiO_2$ films can be deposited at low substrate temperatures of about 300 °C without extra plasma enhancement [33,34]. However, very few studies were reported on this technique. Chang et al. fabricated N-doped $TiO_2$ films by AIP and found that such doping generally reduced photocatalytic efficacy, but the performance increased at nitrogen partial pressures of 5~10% in oxygen [34]. Zhang et al. synthesized composite $TiO_xN_y$ films

with pulsed bias AIP in a gaseous mixture of Ar, $N_2$, and $O_2$ [35]. At a substrate bias of about 100 V, these films exhibited a mixture of anatase and rutile phases with O elements partly replaced by N elements, and the threshold wavelength of optical absorption shifted from 367 to 400 nm after doping due to bandgap narrowing [35].

In this study, N-doped $TiO_2$ thin films were prepared using AIP without substrate heating and/or applied bias. This not only makes the fabrication process much easier but also allows various possible substrates to be used. In a gas mixture of $N_2$ and $O_2$, the partial pressure of $N_2$ was adjusted to attain different $N_2/O_2$ ratios. The surface structures, crystallinities, absorption spectra, optical properties, and bandgaps of these films were characterized by simulations and experiments. The results indicated that the bandgap was the narrowest when $P_{N2} = 3 \times 10^{-3}$ torr and $P_{N2}/P_{O2} = 1/4$, and the methylene blue test verified the corresponding enhanced photocatalytic performance. It is well explained by the sample's prominent Ti–N peaks observed in the XPS spectra.

## 2. Materials and Methods

### 2.1. Sample Preparation

B270 glass slides and silicon wafers were used as substrates for the deposition of N-doped $TiO_2$ thin films using AIP. Glass substrates were first polished with wet cotton moistened by $CeO_2$ powders to increase the adhesion of deposited layers, washed with deionized water, and then blown with clean nitrogen gas. Silicon substrates were cleaned with alcohol and then dried. The AIP system consisted of the arc power supply, vacuum chamber, and gas exhaust pump. The experimental parameters are as follows: purity of the 2-inch Ti target: 99.99%; arc power supply voltage: 50 V; arc power supply current: 75 A; working gases: $O_2$ (purity > 99.995%) and $N_2$ (purity > 99.995%); substrate temperature: room temperature; deposition time: 120 s; base pressure: $1 \times 10^{-5}$ torr; working pressure: $1.5 \times 10^{-2}$ torr. Samples were prepared under different $N_2$ partial pressure ($P_{N2}$) and $O_2$ partial pressure ($P_{O2}$), as listed in Table 1.

**Table 1.** N-doped $TiO_2$ thin films prepared by using AIP under different $N_2$ and $O_2$ partial pressures.

| Sample | Substrate | N$_2$ Partial Pressure, P$_{N2}$ ($\times 10^{-3}$ torr) | O$_2$ partial pressure, P$_{O2}$ ($\times 10^{-3}$ torr) | P$_{N2}$/P$_{O2}$ |
|---|---|---|---|---|
| TiO$_2$ | Silicon | 0 | 15.0 | 0 |
| TiON-n150 | Silicon | 1.5 | 13.5 | 1/9 |
| TiON-n220 | Silicon | 2.2 | 12.8 | 1/5.8 |
| TiON-n300 | Silicon | 3.0 | 12.0 | 1/4 |
| TiON-n440 | Silicon | 4.4 | 10.6 | 1/2.4 |
| TiON-n600 | Silicon | 6.0 | 9.0 | 1/1.5 |

### 2.2. Sample Characterization

The transmittance of samples on glass substrates was examined with the Varian Cary 5E spectrometer (Varian, Palo Alto, CA, USA) at wavelengths ($\lambda$) of 250~700 nm in the UV to VIS range. For samples on silicon substrates, their surface morphology was characterized by scanning electron microscope (SEM) (S-4800, HITACHI, Krefeld, Germany) and atomic force microscope (AFM) (Digital Instruments, Bresso, Italy). Their crystallinity and composition were analyzed with X-ray diffraction (XRD) (MultiFlex, RIGAKU, Tokyo, Japan) and X-ray photoelectron spectroscopy (XPS) (ESCA PHI 1600, ULVAC-PHI, Kanagawa, Japan), respectively. Optical properties, including refractive index (*n*) and extinction coefficient (*k*) together with film thickness (*d*), were measured by the VASE M44 ellipsometer (J.A. Woollam Co., Lincoln, NE, USA). The deposited film on the sample's surface was irradiated by incident light from 350 to 700 nm at incident angles of 50°, 55°, and 60°. The $\Psi$ and $\Delta$ values of the deposited film were obtained experimentally by the ellipsometer. These values were related to the Fresnel reflection coefficients for the *p*- and *s*-polarized light, $R_p$

and $R_s$, respectively, as $\tan(\Psi)e^{i\Delta} = R_p/R_s$. The built-in Cauchy model was then used to fit the data with the following equations:

$$n(\lambda) = A + B/\lambda^2 + C/\lambda^4, \text{ and } k(\lambda) = De^{E(12400(1/\lambda - 1/F))}, \tag{1}$$

where $A \sim F$ are fitting parameters. With measured ($\Psi_{exp}$ and $\Delta_{exp}$) and modeled ($\Psi_{model}$ and $\Delta_{model}$) values, the mean squared error (MSE) can be calculated to indicate the quality of the estimator:

$$\text{MSE} = \sqrt{\frac{1}{2N-M}\sum_{i=1}^{N}\left[\left(\frac{\Psi_i^{model} - \Psi_i^{exp}}{\sigma_{\Psi,i}^{exp}}\right)^2 + \left(\frac{\Delta_i^{model} - \Delta_i^{exp}}{\sigma_{\Delta,i}^{exp}}\right)^2\right]}, \tag{2}$$

where $N$ is the number of measured ($\Psi$, $\Delta$) pairs, $M$ is the number of fitting parameters, and $\sigma$ is the standard deviation of experimentally derived $\Psi$ or $\Delta$ [26]. Moreover, since the surface roughness of AIP-deposited films is large, another built-in model, the fitted surface roughness layer on the film's top, was added to the Cauchy model. This roughness layer is treated as an effective medium approximation (EMA) of 50% material and 50% void [36].

In addition to optical properties, the bandgap energy of these films can also be derived from the Tauc-Lorentz (TL) model of the generalized oscillator layer model built into the ellipsometer. The imaginary part of the dielectric function ($\varepsilon_2 = 2nk$) at a photon energy of $E$ can be expressed as:

$$\varepsilon_2 = \frac{A(E - E_g)^2 CE_0}{(E^2 - E_0^2)^2 + C^2E^2} \text{ for } E > E_g \text{ and } \varepsilon_2 = 0 \text{ for } E \leq E_g, \tag{3}$$

where the amplitude of the oscillator $A$, the broadening parameter $C$, the peak in the joint density of states $E_0$, and the bandgap energy $E_g$ are four parameters to be fitted [26,37].

The photocatalytic performance of samples was evaluated by the degradation of methylene blue (MB), a prevalent test pollutant in photocatalysis [38,39], under lights of specific wavelengths. The Beer–Lambert law states that the absorbance ($A$) of a material to a specific light is proportional to the absorption coefficient of the material ($\alpha$), the path length of the light ($x$), and the concentration of the material ($c$): $A = \alpha xc$ [40,41]. This absorbance is related to the transmittance ($T$) of the light to the same material as $A = -\log T$. Therefore, as the photocatalytic efficiency increases, more methylene blue molecules are decomposed, leading to a decrease in their concentration. This in turn reduces the absorbance of the light. To perform photocatalytic tests, TiO$_2$ samples of 20 mg were added to 100 mL of 10 mg/L of MB solution inside a cuvette. After exposure to light of different wavelengths (420, 440, 460, 480, and 500 nm) emitted by a photoluminescence system for 30 min, the absorbance was measured with the Varian Cary 5E spectrometer. The difference in the absorbance ($\Delta$A) before and after the photocatalysis was calculated. The higher value of $\Delta$A indicates more MB molecules decomposed, the better photocatalytic performance.

## 3. Results and Discussion

### 3.1. XRD and Surface Morphologies of N-Doped TiO$_2$ Films

In the present study, TiO$_2$ films were prepared using AIP without substrate heating. As reported, the crystal phases of TiO$_2$ are anatase and rutile at 400 and 700 °C, respectively [13]. Figure 1a shows the XRD patterns of pure TiO$_2$ films on silicon and B270 glass substrates and N-doped TiO$_2$ films on silicon substrates. Clearly, all samples exhibit amorphous structures, as suggested by Karunagaran et al. that the structures of TiO$_2$ films deposited at ambient temperature were amorphous [42]. Another possible reason is that these films are so thin (thickness ~200 nm, as discussed below) that the granular sizes are too small to form observable crystal structures. In our other study, the TiO$_2$ film was deposited on a PC plastic substrate using AIP without substrate heating. The TiO$_2$ film illustrates the anatase crystal structure, where the XRD peak locates at 2θ of about

25.271 degrees (Pattern COD 5000223), as shown in Figure 1b. The peak's FWHM (Full width at half maximum) is apparently more significant than those deposited on silicon wafers and glass substrates with a substrate temperature of 250 °C. The grand size, evaluated by the Scherrer equation, of $TiO_2$ without substrate heating is smaller than that with substrate heating. Substrate temperature does affect the growth of $TiO_2$ crystals during AIP deposition. Nevertheless, these films fabricated without substrate heating still work as good photocatalysts in this study.

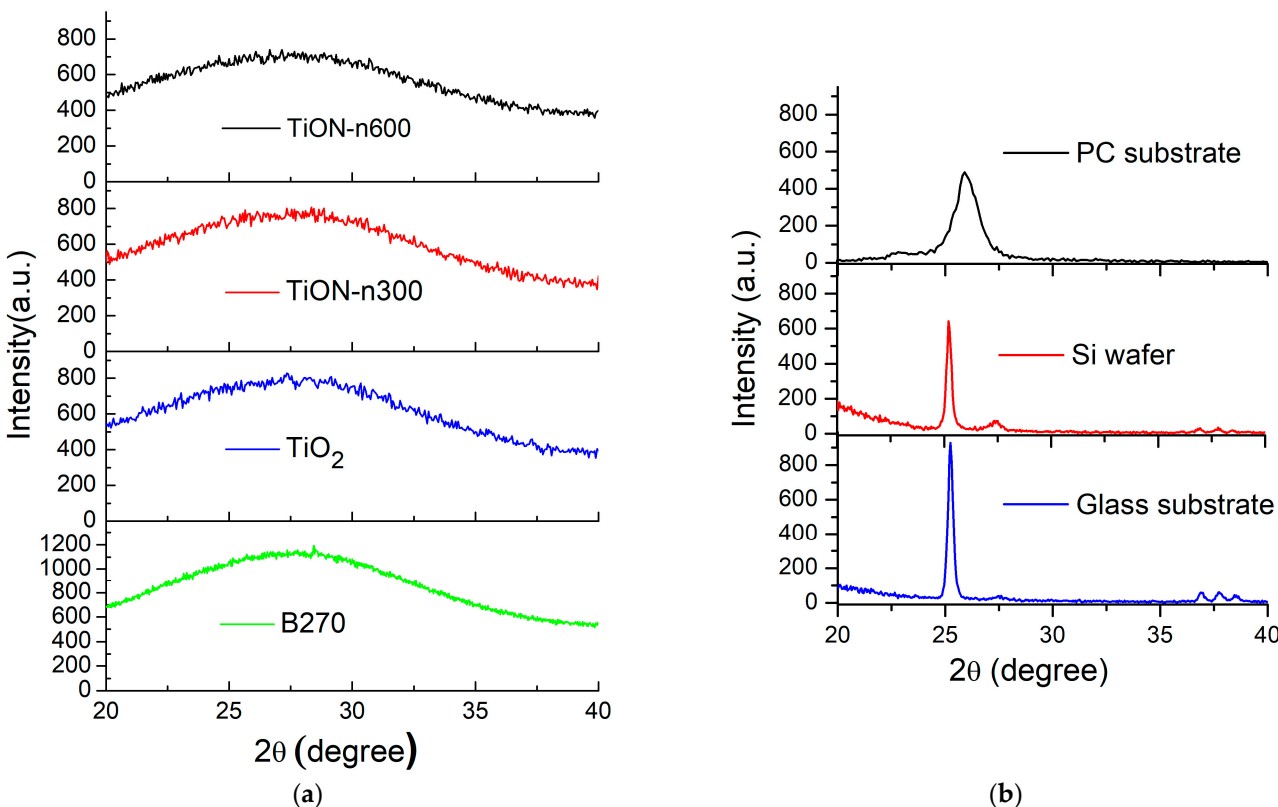

(a)                                                                                      (b)

**Figure 1.** (**a**) XRD patterns of samples TiON-n600, TiON-n300, $TiO_2$, and B270, (**b**) XRD patterns of samples with 2-μm $TiO_2$ deposited on a PC substrate without substrate heating, a silicon wafer and a glass substrate both with substrate temperature 250 °C.

### 3.2. Surface Morphology of N-Doped $TiO_2$ Films

The surface morphology of the deposited $TiO_2$ film was characterized by AFM and SEM. The surface roughness of the AIP-deposited film shown in Figure 2a was larger than that of a film prepared with sputtering [26]. Tiny particles with sizes ranging from 0.3–0.8 μm are distributed on the surface, and the maximum particle size is about 4 μm, as shown in Figure 2b. Moreover, the samples have many tiny particles in the AIP-deposited film. The surface area of the resulting particle is more significant than the film fabricated by the other vacuum deposited method. That can strengthen the photocatalysis effect.

### 3.3. Transmittance of N-Doped $TiO_2$ Films

The transmittance of samples on glass substrates was measured with the Varian Cary 5E spectrometer. For films on silicon substrates, the ellipsometer was used to measure the $\Psi$ and $\Delta$ values for fitting their optical properties, including *n*, *k*, and thickness *d*. Figure 3 shows the transmittance in the 250~700 nm region for all samples in Table 1. With a deposition time of 120 s, the thicknesses of samples $TiO_2$, TiON-n150, TiON-n220, TiON-n300, TiON-n440, and TiON-n600 are about 191, 187, 176, 197, 197, and 202 nm, respectively. The transmittance for all samples in the second peak in the VIS region is higher than 85%. Moreover, sample TiON-n300 (pink color) has good transmittance at 360 and 500 nm. The

threshold wavelengths of all samples are around 315 nm. The different spectrum shifting, resulting from the variation of the deposited film's optical thickness ($n \times d$), is discussed in the following section.

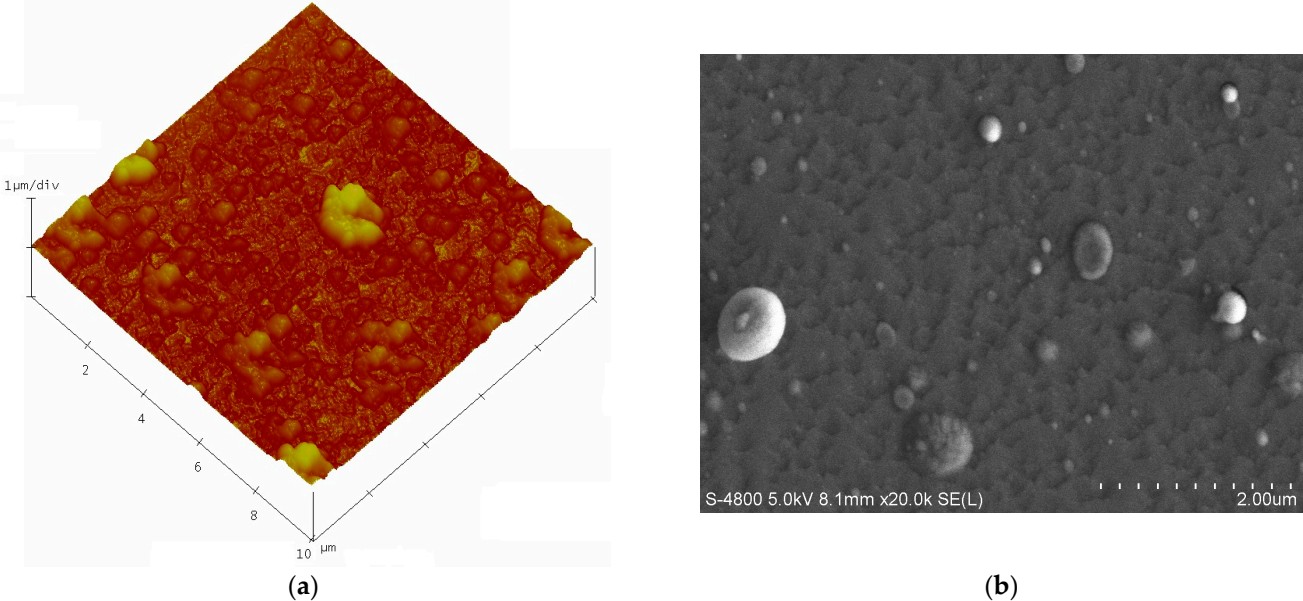

(**a**)                                                                 (**b**)

**Figure 2.** Surface morphology of sample TiO$_2$ characterized by (**a**) AFM and (**b**) SEM.

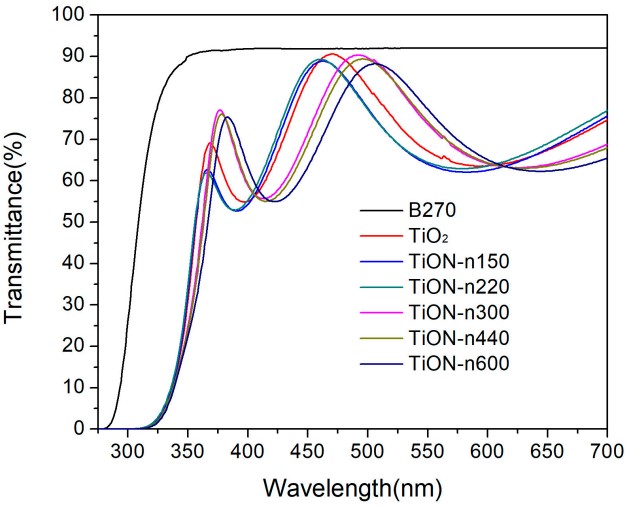

**Figure 3.** Transmittance in the 250~700 nm region for all samples listed in Table 1.

### 3.4. Optical Properties of N-Doped TiO$_2$ Films

The VASE M44 ellipsometer with built-in simulation models was used to derive the optical properties of N-doped TiO$_2$ films. For example, Figure 4 shows the fitted results of the deposited TiO$_2$ film on a silicon substrate by the Cauchy model. Measured and fitted Ψ values are shown as dashed and solid lines with an MSE value of 1.98 for three different incident angles of 50°, 55°, and 60°. The fitted thickness of the TiO$_2$ film is 188 nm, and the top surface roughness layer is fitted to be 3.76 nm due to its rough surface mentioned in Section 3.2.

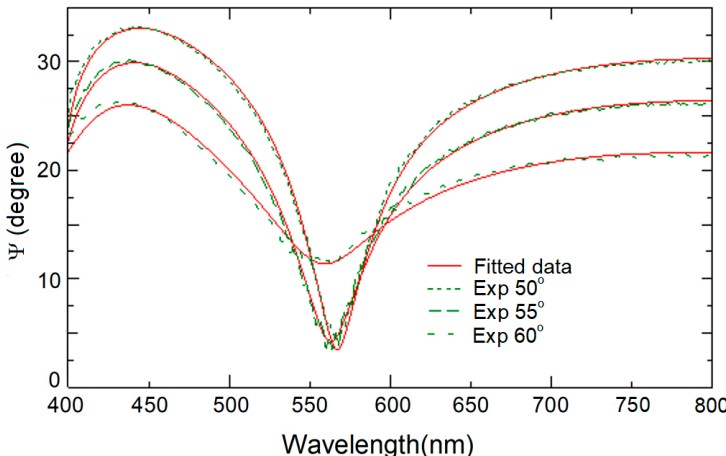

**Figure 4.** Measured (dashed lines) and Cauchy model-fitted $\Psi$ (solid lines) values for three different incident angles of $50°$, $55°$, and $60°$ are shown for the TiO$_2$ film deposited on a silicon substrate.

Figure 5a shows the films' fitted surface roughness and thickness under different $P_{N2}$. The thickness generally increases with increasing pressure due to crystal grain growth. The TiON-n220 sample deposited at a $P_{N2}$ of $2.2 \times 10^{-3}$ torr has the smallest fitted surface roughness, as it may have the densest grown grains. The refractive index ($n$) and extinction coefficient ($k$) at 550-nm wavelength are plotted against the N$_2$ partial pressure in Figure 5b. The refractive index approximately increases with increasing N$_2$ pressure, being 2.3988, 2.4247, 2.4334, 2.446, and 2.4826 at $P_{N2}$ of 0, $1.5 \times 10^{-3}$, $2.2 \times 10^{-3}$, $3.0 \times 10^{-3}$, $4.4 \times 10^{-3}$, and $6.0 \times 10^{-3}$ torr, respectively. As mentioned in Section 3.1, the grain growth due to the increased deposited thickness may increase the refractive index by reducing grain boundaries and voids in the film. However, no apparent trend exists between the Cauchy model-fitted extinction coefficient ($k$) and the $P_{N2}$.

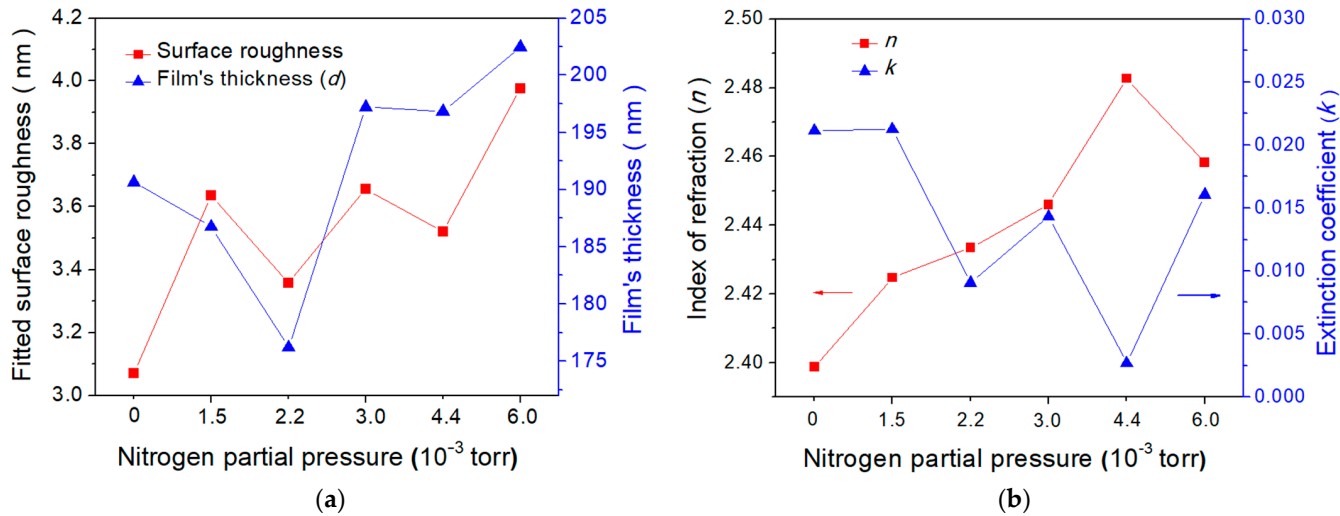

**Figure 5.** (**a**) Fitted surface roughness and thickness of these films under different N$_2$ partial pressures. (**b**) Refractive index ($n$) and extinction coefficient ($k$) at 550-nm wavelength under different N$_2$ partial pressures.

### 3.5. Bandgap Energy of N-Doped TiO$_2$ Films

To increase the photocatalytic efficiency of TiO$_2$, the optical bandgap can be reduced by N-doping. With this doping, the O vacancy state serves as an efficient generation-recombination center, and the N 2p-doped state above the valence band mixes with the O 2p valence band to narrow the bandgap [43]. However, the absorption bandgaps of nitrogen-doped TiO$_2$ films, which have polycrystalline structures, are difficult to be determined

due to the smooth transmittance curves shown in Figure 3 at the threshold wavelength of 315 nm. For the same reason, their extinction coefficients ($k$) fitted by the Cauchy model in Section 3.4 are somewhat disorderly with increasing $N_2$ partial pressure, as shown in Figure 5a. The $k$ values may also be disorderedly transformed into absorption coefficients by $\alpha = 4\pi k/\lambda$ equation and energy gaps.

Jellison and Modine developed the TL model in 1996. It models the dielectric function of many different amorphous materials and measures its energy gap well [44]. In this study, the N-doped $TiO_2$ films are similar to amorphous ones due to their multi-crystalline structures of tiny crystalline grains, as mentioned in Section 3.1. Therefore, the measured $\Psi$ values of all samples were fitted again with the TL model. For three different incident angles of 50°, 55°, and 60°, the $\Psi$ values fitted by the TL model are somewhat like those of $\Psi$ (solid line) fitted by the Cauchy model shown in Figure 4.

The energy gaps $E_g$s of all samples were derived from the TL model of the generalized oscillator layer model shown in Equation (3) and plotted in Figure 6. The bandgap of the AIP-fabricated film without doping is about 3.18 eV, similar to the anatase $TiO_2$. The energy gap decreases with increasing $N_2$ partial pressure to $2.2 \times 10^{-3}$ torr, being 3.15, 3.12, and 3.09 eV for $P_{N2}/P_{O2} = 1/9$, 1/5.8, and 1/4, respectively. However, with further increased $N_2$ partial pressure, the gap energy increases again to 3.1 and 3.13 eV for $P_{N2}/P_{O2} = 1/2.4$ and 1/1.5, respectively. The red shift was observed with increasing $N_2$ partial pressure from 0 to $3 \times 10^{-3}$ torr (TiON-n300). Yang et al. applied first-principles density functional theory to study N-doped $TiO_2$ crystals with various nitrogen concentrations and found that the red shift observed in the experiments was related to the locations of N 2p states [45]. With the sol–gel method, Cheng et al. synthesized N-doped $TiO_2$ nano-photocatalysts in the presence of ammonium chloride. After N-doping, the light absorption edge red-shifted to the VIS region, enhancing VIS-light photocatalytic activity [46]. The red shift phenomenon was also observed in other doped $TiO_2$ [47–49]. The results suggest that the N-doped $TiO_2$ film TiON-n300 fabricated by AIP in a gaseous mixture of $N_2$ and $O_2$ with $P_{N2}/P_{O2} = 1/4$ could achieve the narrowest 3.09 eV bandgap and highest absorbance in the VIS region of around 400.7 nm.

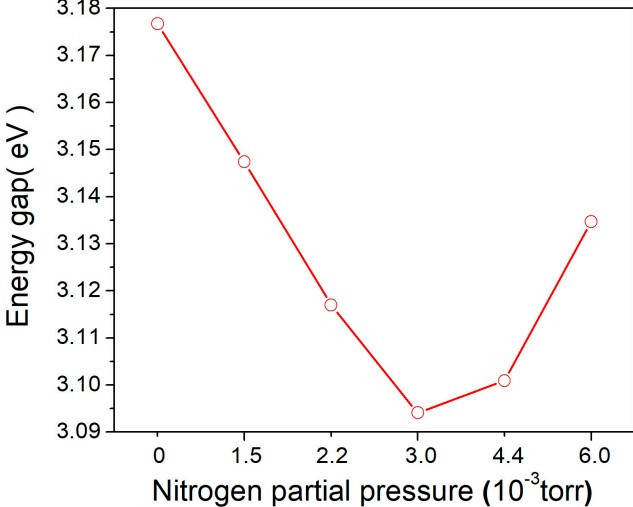

**Figure 6.** Energy gaps of N-doped $TiO_2$ films under different $N_2$ partial pressures.

The absorbance spectra derived from the TL-mode are shown in Figure 7, indicating that compared with other samples, the spectrum of the TiON-n300 sample shifted obviously to the VIS region. These results suggest that the N-doped $TiO_2$ film fabricated by AIP in a gaseous mixture of $N_2$ and $O_2$ with $P_{N2}/P_{O2} = 1/4$ can attain the narrowest bandgap and the highest absorbance in the VIS region.

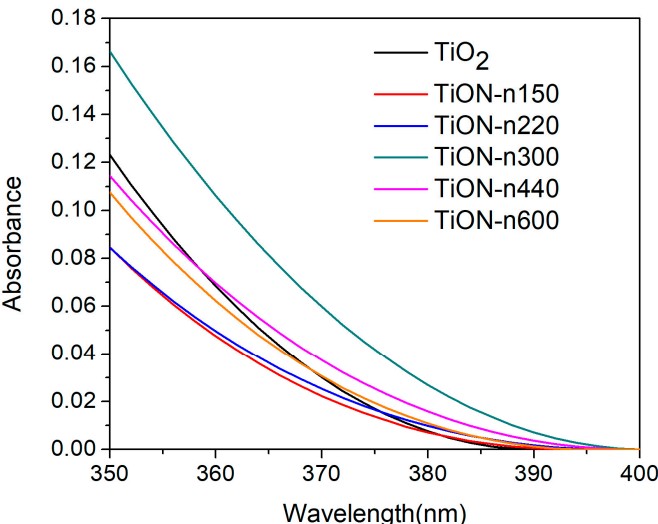

**Figure 7.** Absorbance spectra of all samples derived from the TL model.

### 3.6. XPS of N-Doped TiO₂ Films

In order to further investigate the phenomenon that the N-doped $TiO_2$ film with $P_{N2}/P_{O2} = 1/4$ exhibits the best photocatalytic performance, XPS of samples TiON-n300 and TiON-n600, deposited with $P_{N2}/P_{O2} = 1/4$, 1/1.5, respectively, were conducted to evaluate their compositions. Figure 8 shows the XPS spectra of oxygen for the samples, with peaks ranging between 528 and 534 eV. Typical peaks near 528.4, 529.8, and 531.5 eV are observed. The strongest 529.8 eV peak corresponds to the O–Ti binding [50], and the much weaker 528.4 eV one represents the low-binding-energy (LBO) oxygen originating from those non-bridging oxygen (NBO) species [44]. Hsu et al. suggested that the bond at the non-bridging sites reduced the total energy of the oxide formation. The concentration of LBO in the $TiO_2$ film was always the smallest among all components [51]. The 531.5 eV peak is from the oxygen absorption of C–O and $H_2O$, which may come from the air and organic pollution [52].

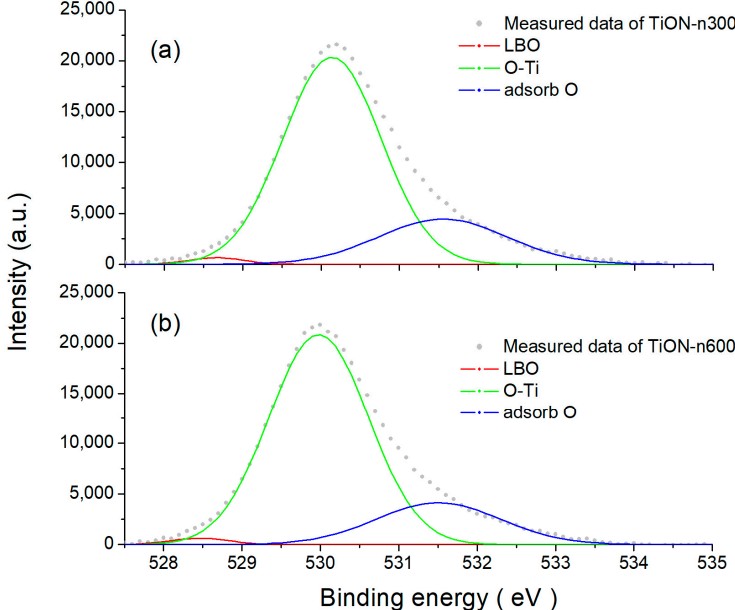

**Figure 8.** XPS spectra of O 1s core line for samples (**a**) TiON-n300 and (**b**) TiON-n600.

The XPS spectra of Ti for both samples are shown in Figure 9. These signals comprise two groups, the stronger one from Ti $2p_{3/2}$ and the weaker one from Ti $2p_{1/2}$. Each group

can be decomposed into three peaks of $Ti^{2+}$, $Ti^{3+}$, and $Ti^{4+}$. For example, in the Ti $2p_{3/2}$ group, the 457 eV and 458.7 eV peaks correspond to the $Ti^{3+}$ and $Ti^{4+}$ bonds. The most apparent $Ti^{4+}$ peak (the blue ones) originates from the normal $TiO_2$. When $O_2$ molecules are less, three O atoms share two Ti atoms, forming $Ti_2O_3$ and the weaker $Ti^{3+}$ peak (the green ones). The $Ti^{3+}$ peak of TiON-n300 shown in Figure 9a is weaker than that of TiON-n600 shown in Figure 9b. It explains that the transmittance of TiON-n300 is better than that of TiON-n600 shown in Figure 3. The 456 eV $Ti^{2+}$ peaks (the red ones) come from the Ti–N bonds [52]. Although the $P_{N2}$ is higher in the TiON-n600 sample, the Ti–N bond is somewhat weaker than that in the TiON-n300 sample. It explains why the TiON-n300 sample exhibits the narrowest energy gap and the best photocatalytic performance.

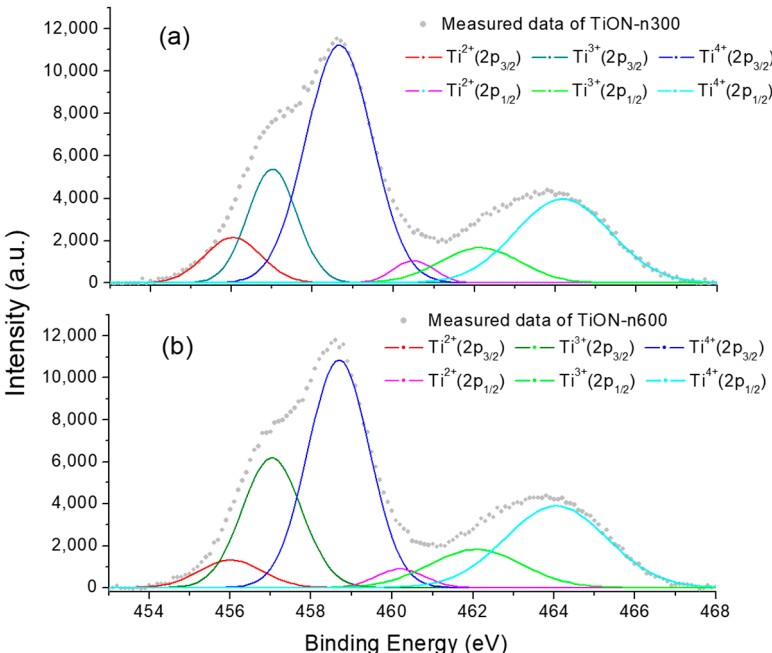

**Figure 9.** XPS spectra of Ti 2p core line for samples (**a**) TiON-n300 and (**b**) TiON-n600.

Figure 10 shows the XPS spectra of N for both samples. The TiON-n300 sample exhibits a clear 396.7 eV peak from the Ti–N bond, as shown in Figure 10a [53]. The intensity of this peak is reduced in the TiON-n600 sample, as shown in Figure 10b. It again verifies that the TiON-n300 sample has the narrowest energy gap and the highest photocatalytic efficiency due to the light absorption of the Ti–N bond.

### 3.7. Photocatalytic Performance of N-Doped $TiO_2$ Films

The degradation of MB is used to evaluate the photocatalytic performance of N-doped $TiO_2$ films. As the photocatalytic efficiency increases, more MB molecules are decomposed, leading to a decrease in their concentration. According to the Beer–Lambert law, this in turn reduces the absorbance to the light measured at 600 nm since MB exhibits the highest absorbance at this wavelength. Figure 11a shows the absorbance of the pure $TiO_2$ sample in the MB solution before and after exposure to 420 nm light. The efficiencies are qualitatively analyzed by calculating the absorbance difference, such as 0.1863% shown in Figure 11a, before and after the MB solution is reacted with the chosen sample under the exposure of varying photocatalytic light sources. Figure 11b shows $\Delta A$, the absorbance difference divided by the peak value of the absorbance spectrum before exposure, of the samples pure $TiO_2$, TiON-n300, and TiON-n600 in percentage for exposure to photocatalytic lights at 420, 440, 460, 480, and 500 nm wavelength, respectively. As increasing the wavelength, the $\Delta A$ value decreases, indicating worse photocatalytic performance. However, the performance of TiON-n600 appears better than that of the pure $TiO_2$ film at excitation wavelengths of 460, 480, and 500 nm. The nitrogen dopant indeed reduces the photocatalytic bandgap. Among

these samples, TiON-n300 shows the largest $\Delta A$ and hence the highest photocatalytic efficiency, verifying that it has the narrowest bandgap.

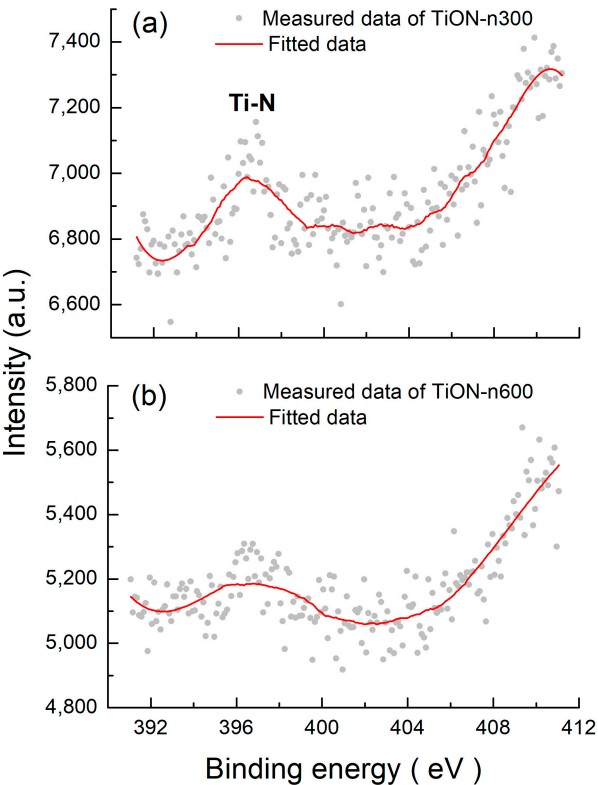

**Figure 10.** XPS spectra of N 1s core line for samples (**a**) TiON-n300 and (**b**) TiON-n600.

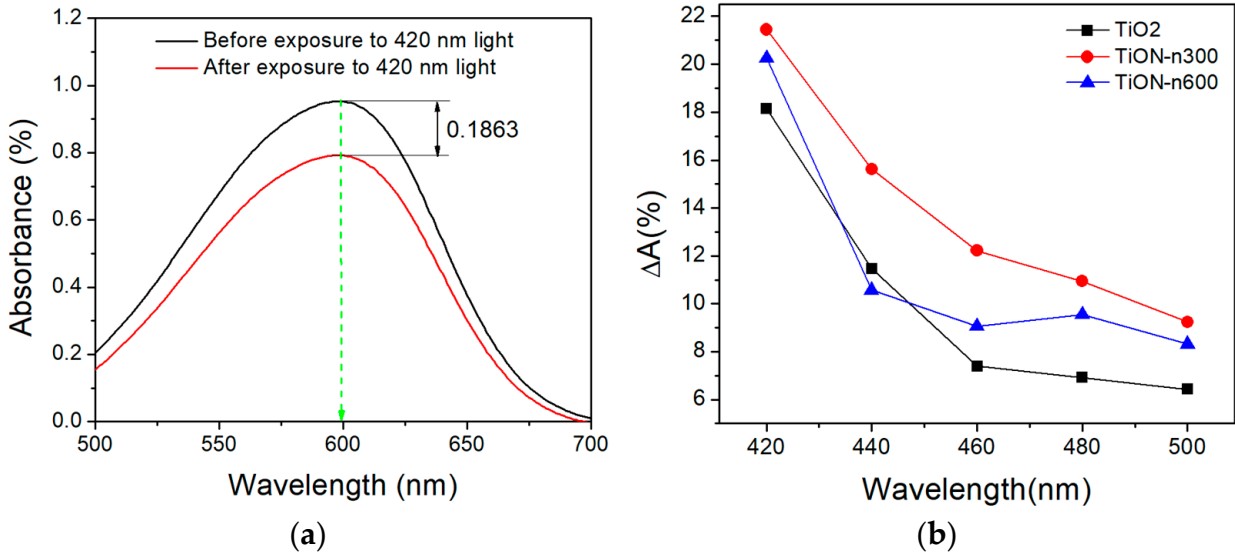

**Figure 11.** (**a**) Absorbance of pure $TiO_2$ sample in the MB solution before and after exposure to 420 nm light. (**b**) Absorbance change rates of different samples in MB solution irradiated with different photon wavelengths.

Table 2 compares nitrogen-doped $TiO_2$ films prepared at different parameters of substrate temperature and negative bias in other studies. In the present work, the samples deposited at no bias and substrate heating exhibited amorphous structures. At $P_{N2}/P_{O2} = 1/4$ condition, the nitrogen-doped $TiO_2$ film showed the best photocatalytic performance.

**Table 2.** A comparison of nitrogen-doped TiO$_2$ films prepared by arc ion plating (AIP).

| Substrate Conditions | Working Pressures | Main Findings | References |
|---|---|---|---|
| 1. No substrate bias<br>2. T$_{sub}$ not specified (No substrate heating) | 1. P$_{total}$ = 0.8 Pa<br>2. P$_{N2}$/P$_{total}$ = 5%~50% | 1. Amount of TiO$_2$ anatase phase decreased with N doping.<br>2. N doping generally reduced photocatalytic efficacy, but the performance increased at P$_{N2}$/P$_{total}$ = 5%~10% | [34] |
| 1. Substrate bias from 0 to −700 V<br>2. T$_{sub}$ not specified | 1. P$_{total}$ = 0.2 Pa<br>2. P$_{N2}$/P$_{O2}$ = 1/2 | 1. At a bias of −100 V, these films exhibited a mixture of anatase and rutile phases. | [35] |
| 2. No substrate bias<br>1. T$_{sub}$ = 200 °C | 1. P$_{total}$ = 1~1.3 Pa<br>2. P$_{N2}$/P$_{O2}$ = 0, 2/1~5/1 | 1. N doping inhibited the anatase growth and promoted rutile formation.<br>2. N-doped TiO$_2$ film not only reduced the UV catalytic performance but also extended the catalytic properties to the sunlight. | [54] |
| 1. Substrate bias from 0 to −600 V<br>2. T$_{sub}$ = 200~300 °C | 1. P$_{total}$ = 1~1.3 Pa<br>2. P$_{N2}$/P$_{O2}$ = 13/4 | 1. Absorption edges of the as-deposited films are increased with the rising of the negative bias, and a maximum of 550 nm is achieved under −600 V bias. | [55] |
| 1. Substrate bias from 0 to −600 V @20 kHz<br>2. T$_{sub}$ = 150~300 °C | 1. P$_{total}$ = 1~1.8 Pa<br>2. P$_{N2}$/P$_{O2}$ = 13/4 | 1. Applying the negative pulse bias extended the absorption edge 5–40 nm towards the visible region. | [56] |
| 1. No substrate bias<br>2. No substrate heating | 1. P$_{total}$ = 2 Pa<br>2. P$_{N2}$/P$_{O2}$ = 0, 1/9~1/1.5 | 1. All samples exhibited amorphous structures.<br>2. Sample with P$_{N2}$/P$_{O2}$ = 1/4 showed the best photocatalytic performance. | The present work |

T$_{sub}$: substrate temperature.

## 4. Conclusions

N-doped TiO$_2$ films were prepared by AIP under various gas mixing ratios without substrate heating and/or applied bias. The N-doping led to a similar amorphous structure of these films with rough surfaces. Analysis of their transmittance, optical properties, and bandgap energies indicated that the N-doped TiO$_2$ film with the P$_{N2}$/P$_{O2}$ ratio of 1/4 could attain the narrowest bandgap and the highest absorbance in the VIS region. This narrowest gap corresponded to the best photocatalytic performance, as verified by the MB test. Moreover, the sample's Ti and N XPS spectra exhibited prominent Ti–N peaks, responsible for its narrowest energy gap and highest photocatalytic efficiency. These results conclude that AIP can prepare the N-doped TiO$_2$ film without substrate heating, and at a certain P$_{N2}$/P$_{O2}$ ratio, the bandgap can be efficiently reduced to enhance its photocatalytic ability. This study further explains the mechanisms of N-doped TiO$_2$ photocatalysts without substrate heating. The result can be considered for the photocatalysts deposition of heat-labile substrates.

**Author Contributions:** Conceptualization, J.-C.H.; methodology, W.-C.H. and J.-C.H.; software, W.-C.H. and G.-Y.Y.; validation, Y.-S.S.; formal analysis, J.-L.W. and G.-Y.Y.; investigation, J.-L.W.; resources, J.-L.W.; data curation, W.-C.H. and H.-Y.W.; writing—original draft preparation, Y.-S.S. and H.-Y.W.; writing—review and editing, H.-Y.W. and J.-C.H.; visualization, J.-C.H.; supervision, Y.-S.S. and J.-C.H.; project administration, W.-C.H. and G.-Y.Y.; funding acquisition, H.-Y.W. All authors have read and agreed to the published version of the manuscript.

**Funding:** This research was funded by the Ministry of Science and Technology of Taiwan, grant numbers MOST 111-2112-M-030-004 and MOST 111-2221-E-030-007.

**Institutional Review Board Statement:** Not applicable.

**Informed Consent Statement:** Not applicable.

**Acknowledgments:** We thank the Ming Chi University of Technology who supported the FESEM, XPS, and XRD experimental work.

**Conflicts of Interest:** The authors declare no conflict of interest.

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
