# Peer review of "Characterization of Nitrogen-Doped TiO2 Films Prepared by Arc Ion Plating without Substrate Heating in Various N2/O2 Gas Mixture Ratios"

_coatings, doi:10.3390/coatings13030654_

Round 1

Reviewer 1 Report

Coatings-2281820

In the manuscript titled “Characterization of Nitrogen-Doped TiO2 Films Prepared by Arc Ion Plating without Substrate Heating in Various N2/O2 Gas Mixture Ratios authors have described the fabrication of Nitrogen-Doped TiO2 films using arc ion plating (AIP) in different ratios of nitrogen partial pressure (PN2) to oxygen partial pressure (PO2) without substrate heating and/or applied bias. They evaluated the transmittance and optical properties of the thin films. The work is adequate and can be published in after minor revisions.

Comments are:

o   Revise the abstract with the addition of the significant findings, and the conclusions reached to your abstract.

o   The introduction should be enriched by citing some recent works.

o   Explain the novelty of the current work at the end of the introduction.

o   Explain with support to the use of AIP fabrication method.

o   The Figure 1 (Unit) should be in same font. Also use the uniform units throughout the article.

o   The resolution of all figures is very low. Revise them.

o   In line 207 different constants have been used. Write names of the constants were used for first time.

o   The photocatalytic test can be more detailed. It is very short and difficult to understand.

o   In photocatalytic test, the authors have claimed that MB exhibits the highest absorbance at 600nm. While the reported wavelength is 664-666nm for highest absorbance for MB.

o   Add a comparative study table containing the similar previous studies.

o   Improve the English grammar and writing skills for this paper.

o   Recent References should be used. (Mostly from 2017 to onwards)  

Author Response

Response Comments and Suggestions

Dear Editor,

Thank you very much for your useful comments and suggestions. To address the referees' concerns, we have made revisions to the manuscript accordingly. All modified parts were highlighted using “Track Changes” in Word. Detailed responses to the referees’ comments are listed below point by point:

Reviewer 1

In the manuscript titled “Characterization of Nitrogen-Doped TiO2 Films Prepared by Arc Ion Plating without Substrate Heating in Various N2/O2 Gas Mixture Ratios” authors have described the fabrication of Nitrogen-Doped TiO2 films using arc ion plating (AIP) in different ratios of nitrogen partial pressure (PN2) to oxygen partial pressure (PO2) without substrate heating and/or applied bias. They evaluated the transmittance and optical properties of the thin films. The work is adequate and can be published in after minor revisions.

Response: The authors thank this reviewer for all helpful comments.

Comments are:

o      Revise the abstract with the addition of the significant findings, and the conclusions reached to your abstract.

Response: We have revised the Abstract accordingly.

o      The introduction should be enriched by citing some recent works.

Response: We have modified the Introduction accordingly to include more recent works.

o      Explain the novelty of the current work at the end of the introduction.

Response: We have added few more sentences at the end of Introduction to address the novelty of the present work.

o      Explain with support to the use of AIP fabrication method.

Response: The use of AIP fabrication method is explained (with supports from few studies) in Introduction as follows:

Among these methods, arc ion plating (AIP) is particularly interesting because anatase TiO2 films can be deposited at low substrate temperatures of about 300 °C without extra plasma enhancement [26,27]. However, very few studies were reported on this technique. Chang et al. fabricated N-doped TiO2 films by AIP and found that such doping generally reduced photocatalytic efficacy, but the performance increased at nitrogen partial pressures of 5 ~ 10% in oxygen [27]. Zhang et al. synthesized composite TiOxNy films with pulsed bias AIP in a gaseous mixture of Ar, N2, and O2 [28]. At a substrate bias of about 100 V, these films exhibited a mixture of anatase and rutile phases with O elements partly replaced by N elements, and the threshold wavelength of optical absorption shifted from 367 to 400 nm after doping due to bandgap narrowing [28].

o      The Figure 1 (Unit) should be in same font. Also use the uniform units throughout the article.

Response: Figure 1 is revised

o     The resolution of all figures is very low. Revise them.

Response: All figures are revised to high resolution.

o      In line 207 different constants have been used. Write names of the constants were used for first time.

Response: These constants are mentioned in Section 2.2 Sample Characterization where they first appear.

o      The photocatalytic test can be more detailed. It is very short and difficult to understand.

Response: We have described the photocatalytic test in more detail in the revised manuscript.

o     In photocatalytic test, the authors have claimed that MB exhibits the highest absorbance at 600nm. While the reported wavelength is 664-666nm for highest absorbance for MB.

Response: We did find that MB in our tested concentration (10 mg/L) exhibits the highest absorbance at 600nm. And, Figure 11(a) is added in the revised manuscript.

o      Add a comparative study table containing the similar previous studies.

Response: “Table 2. A comparison of nitrogen-doped TiO2 films prepared by arc ion plating (AIP). “ is added the manuscript.

o      Improve the English grammar and writing skills for this paper.

Response: We have thoroughly checked and modified the manuscript to avoid any grammar errors and typos.

o      Recent References should be used. (Mostly from 2017 to onwards)

Response: We have modified the manuscript accordingly to include more recent works.

Reviewer 2 Report

The manuscript entitled "Characterization of Nitrogen-Doped TiO2 Films Prepared by Arc Ion Plating without Substrate Heating in Various N2/O2 Gas Mixture Ratios" presents the results of a studies on the photocatalytic properties of nitrogen-doped TiO2 films. The results of the presented studies are significant in that neither substrate heating nor potential difference was used in the deposition of the layers and still photocatalytic preformance is good. This approach allows a significant broadening of the range of possible substrates to be used. It also makes it possible to significantly reduce the cost of the deposition process and thus the production of photocatalysts. The only two things are missing from this work, in the reviewer's opinion:

1. an emphasis on the importance of not heating the substrate/potential difference in the film deposition process.

2. recycled photocatalytic tests to confirm films stability.

After appropriate correction I recommend acceptance of the paper for publication.

Author Response

Response Comments and Suggestions

Reviewer 2

The manuscript entitled "Characterization of Nitrogen-Doped TiO2 Films Prepared by Arc Ion Plating without Substrate Heating in Various N2/O2 Gas Mixture Ratios" presents the results of a studies on the photocatalytic properties of nitrogen-doped TiO2 films. The results of the presented studies are significant in that neither substrate heating nor potential difference was used in the deposition of the layers and still photocatalytic preformance is good. This approach allows a significant broadening of the range of possible substrates to be used. It also makes it possible to significantly reduce the cost of the deposition process and thus the production of photocatalysts. The only two things are missing from this work, in the reviewer's opinion:

Response: The authors thank this reviewer for all helpful comments.

  1. an emphasis on the importance of not heating the substrate/potential difference in the film deposition process.

Response: We have modified the manuscript accordingly to emphasize the importance of not heating the substrate/potential difference in the film deposition process.

  1. recycled photocatalytic tests to confirm films stability.

Response: The recyclability/reusability of the nitrogen-doped TiO2 films will be investigated in our following experiments.

After appropriate correction I recommend acceptance of the paper for publication.

Reviewer 3 Report

The article, “Characterization of Nitrogen-Doped TiO2 Films Prepared by Arc Ion Plating without Substrate Heating in Various N2/O2 Gas Mixture Ratios” is written well and demonstrated the doping of N atom in TiO2 to increase the light harvesting application of TiO2. Moreover, the authors also conducted a preliminary photocatalytic activity test of the developed materials for the degradation of MB. Below are some crucial suggestions, which may be helpful in increasing the standard of this ms.

1.       How it was confirmed that the odd atom replaced the oxygen of TiO2?

2.       There is plethora of studies on the doping of Nitrogen in TiO2, how this research is different from those studies. A note on the same should be added at the end of the Introduction section. The novelty of the work should be clearly mentioned.

3.       What is the photocatalytic activity of the developed material?

4.       What is the quantum yield of the developed material?

5.       As per SEM image, the size distribution of the developed nanomaterials is too broad, roughly 0.1 to 0.8 um. I suggest to add justification for the broad size distribution. In addition, have authors tried to reduce the size distribution of the particles? Authors may also perform DLS and zeta potential studies to verify the size distribution.

6.       What was the amount of MB adsorbed on the surface of photocatalyst? How much dye was left after putting the dye with photocatalyst under dark conditions for sufficient time duration?

7.       Why the absorbance of TiO2 was decreased after the exposure with 420 nm light? For how much duration TiO2 was exposed to 420 nm light?

8.       A discussion related to the performance’s comparison of developed photocatalysts is missing. The photocatalytic efficiency and quantum yield of all the materials should be compared and added in the revised manuscript. Moreover, the performance of the best material should also be compared with the existing latest researches (in tabular form) to show the superiority of the current work.

Author Response

Response Comments and Suggestions for Reviewer 3

Comments and Suggestions for Authors

The article, "Characterization of Nitrogen-Doped TiO2 Films Prepared by Arc Ion Plating without Substrate Heating in Various N2/O2 Gas Mixture Ratios" is written well and demonstrated the doping of N atom in TiO2 to increase the light harvesting application of TiO2. Moreover, the authors also conducted a preliminary photocatalytic activity test of the developed materials for the degradation of MB. Below are some crucial suggestions, which may be helpful in increasing the standard of this ms.

Response: We thank the reviewer for all the valuable comments and suggestions.

  1. How it was confirmed that the odd atom replaced the oxygen of TiO2?

Response: From Figure 10, the XPS spectra of the N 1s core line for samples showed a clear Ti-N peak in sample TiON-n300. Therefore, we supposed that the N atoms replaced the O atoms of TiO2.

  1. There is plethora of studies on the doping of Nitrogen in TiO2, how this research is different from those studies. A note on the same should be added at the end of the Introduction section. The novelty of the work should be clearly mentioned.

Response: We did address the difference between the present and other studies on N-doped TiO2. We added a few more sentences to the Introduction section to clarify this.

The advantages of applying AIP in fabrication:

Among these methods, arc ion plating (AIP) is particularly interesting because anatase TiO2 films can be deposited at low substrate temperatures of about 300 °C without extra plasma enhancement.

The novelty of the present work:

In this study, N-doped TiO2 thin films were prepared using AIP without substrate heating and/or applied bias. This not only makes the fabrication process much easier but also allows various possible substrates to be used.

  1. What is the photocatalytic activity of the developed material?

Response: The photocatalytic activity of the developed material was verified with the MB test (Section 3.7), suggesting that at a certain PN2/PO2 ratio, the bandgap can be efficiently reduced to enhance its photocatalytic ability.

  1. What is the quantum yield of the developed material?

Response: The quantum yield of the developed materials is a perfect quantitative indicator, which needs to accurately calculate the absolute optical radiation for the measured light and the detector and the quality of the irradiated TiO2 atoms. But, we only need to compare the different photocatalytic activity of the TiO2 film deposited in the various PN2/PO2 ratio.

  1. As per SEM image, the size distribution of the developed nanomaterials is too broad, roughly 0.1 to 0.8 um. I suggest to add justification for the broad size distribution. In addition, have authors tried to reduce the size distribution of the particles? Authors may also perform DLS and zeta potential studies to verify the size distribution.

Response: The few numbers of the particle in the SEM image can not be a statistical distribution. The AIP deposition method cannot control the size distribution except by a filtered arc deposition, which is another issue.   

  1. What was the amount of MB adsorbed on the surface of photocatalyst? How much dye was left after putting the dye with photocatalyst under dark conditions for sufficient time duration?

Response: We could not directly measure how much dye was left after the photocatalytic reaction (it was difficult to measure the concentration/mass of dye). Therefore, we indirectly measure the difference in the absorbance (ΔA) before and after the photocatalysis. According to the Beer-Lambert law (Section 2.2), a higher value of ΔA indicates that more MB molecules are decomposed; hence, the photocatalytic performance is better. It is a standard way of evaluating the performance of a photocatalyst.

  1. Why the absorbance of TiO2 was decreased after the exposure with 420 nm light? For how much duration TiO2 was exposed to 420 nm light?

Response: 1. In brief, the absorbance decreased because MB molecules were decomposed, resulting in a decrease in its concentration.

The Beer-Lambert law states that the absorbance (A) of a material to a specific light is proportional to the absorption coefficient of the material (α), the path length of the light (x), and the concentration of the material (c): A = αxc. A decreased as c decreased.

  1. The exposure duration is 30 min, as stated in Section 2.2:

As stated in Section 2.2, TiO2 samples of 20 mg were added to 100 mL of 10 mg/L of MB solution inside a cuvette. After exposure to light of different wavelengths (420, 440, 460, 480, and 500 nm) emitted by a photoluminescence system for 30 min, the absorbance was measured with the Varian Cary 5E spectrometer. The difference in the absorbance (ΔA) before and after the photocatalysis was calculated. The larger the ΔA value, the more MB molecules decomposed and the better the photocatalytic performance.

  1. A discussion related to the performance's comparison of developed photocatalysts is missing. The photocatalytic efficiency and quantum yield of all the materials should be compared and added in the revised manuscript. Moreover, the performance of the best material should also be compared with the existing latest researches (in tabular form) to show the superiority of the current work.

Response: As stated in the Introduction section, very few studies were reported on fabricating nitrogen-doped TiO2 films using arc ion plating (AIP). We have added a table (see Table 2 in the revision) to compare different N-doped TiO2 films prepared by AIP. However, not all of them investigated the materials' photocatalytic efficiency and quantum yield. Some of them just studied the structures after doping. Although there are countless studies on N-doped TiO2 films fabricated with other techniques such as sol-gel, different kinds of sputtering, different kinds of vapor deposition, electrochemical deposition, electrophoretic deposition, pulsed laser deposition, and hydrothermal synthesis, it is not fair/necessary to make a comparison among all of them. So we just focus on AIP-based N-doped TiO2 films.

Round 2

Reviewer 3 Report

If possible, the authors may provide photocatalytic efficiency in % form.

If not possible, the article can be accepted in current form.

Author Response

Response Comments and Suggestions for Reviewer 3

Response: We thank the reviewer for the valuable suggestions. We have changed Figure 11(b) and the title: Absorbance change rates of different samples in MB solution irradiated with different lights.

And we have defined the ΔA  in Line 344, “ the absorbance difference divided by the peak value of the light before exposure,”